# Host/*Malassezia* Interaction: A Quantitative, Non-Invasive Method Profiling Oxylipin Production Associates Human Skin Eicosanoids with *Malassezia*

**DOI:** 10.3390/metabo11100700

**Published:** 2021-10-13

**Authors:** Yohannes Abere Ambaw, Martin P. Pagac, Antony S. Irudayaswamy, Manfred Raida, Anne K. Bendt, Federico T. Torta, Markus R. Wenk, Thomas L. Dawson

**Affiliations:** 1Precision Medicine Translational Research Programme, Department of Biochemistry, Yong Loo Lin School of Medicine, National University of Singapore, Singapore 119077, Singapore; yambaw@hsph.harvard.edu (Y.A.A.); bchfdtt@nus.edu.sg (F.T.T.); bchmrw@nus.edu.sg (M.R.W.); 2Singapore Lipidomics Incubator, Life Sciences Institute, National University of Singapore, Singapore 119077, Singapore; manfred.raida@newtoncon.de (M.R.); anne.bendt@nus.edu.sg (A.K.B.); 3Department of Molecular Metabolism, Harvard T.H. Chan School of Public Health, Harvard University, Cambridge, MA 02138, USA; 4A*STAR Skin Research Labs (A*SRL), Agency for Science, Technology and Research (A*STAR), Singapore 138648, Singapore; Martin_pagac@asrl.a-star.edu.sg (M.P.P.); antony.irudayaswamy@asrl.a-star.edu.sg (A.S.I.); 5Center for Cell Death, Injury & Regeneration, Departments of Drug Discovery & Biomedical Sciences and Biochemistry & Molecular Biology, Medical University of South Carolina, Charleston, SC 29425, USA

**Keywords:** oxylipins, eicosanoids, lipidomics, skin, non-invasive, mycobiome, *Malassezia*

## Abstract

*Malassezia* are common components of human skin, and as the dominant human skin eukaryotic microbe, they take part in complex microbe–host interactions. Other phylogenetically related fungi (including within *Ustilagomycotina*) communicate with their plant host through bioactive oxygenated polyunsaturated fatty acids, generally known as oxylipins, by regulating the plant immune system to increase their virulence. Oxylipins are similar in structure and function to human eicosanoids, which modulate the human immune system. This study reports the development of a highly sensitive mass-spectrometry-based method to capture and quantify bioactive oxygenated polyunsaturated fatty acids from the human skin surface and in vitro *Malassezia* cultures. It confirms that *Malassezia* are capable of synthesizing eicosanoid-like lipid mediators in vitro in a species dependent manner, many of which are found on human skin. This method enables sensitive identification and quantification of bioactive lipid mediators from human skin that may be derived from metabolic pathways shared between skin and its microbial residents. This enables better cross-disciplinary and detailed studies to dissect the interaction between *Malassezia* and human skin, and to identify potential intervention points to promote or abrogate inflammation and to improve human skin health.

## 1. Introduction

Skin, the largest human organ, fulfils a number of vital functions related to communication with or protection from our environment. The skin has for many decades been calculated to have a surface area of about 2–3 m^2^. However, this does not take into account the complex three-dimensional structure of skin, and if one includes the high level of topography, numerous pores, secretory glands, and hair follicle openings, one finds that the skin surface area is more likely approximates 30 m^2^, much more similar to the lung and gut [1]. It is well known that the microbial communities, or “microbiomes”, of the lung and gut are in constant communication with the human host and are involved in diverse facets of health from digestion to mood. There has also been increasing awareness of the complexity and role of the skin microbiome in human health [2]. As new tools and data become available to study the human skin microbiome, it has become clear that the fungal “mycobiome” plays a larger role in the skin microbiome than in either gut or lung [3]. This may be due to the nature and exposure of skin, to the more aerobic nature of the niche, or to the lower temperature (as fungi are less frequent inhabitants of 37 °C environments).

Appreciation of the fungal “mycobiome” component of microbiomes is also increasing. For instance, in the skin, metagenomic surveys indicate fungi constitute less than 10% [4] of microbial genomes. However, as *Malassezia* yeasts, the vast majority of fungi present on human skin, are 8–10 µm in diameter versus the much smaller typical skin bacteria at 0.8–1 µm, it is very likely that the active fungal biomass represents at least a significant percentage of the functional skin microbial community [5]. It has also recently been demonstrated by modelling community interactions that only inclusion of host, bacterial, and fungal components can generate useful predictions [6].

While *Malassezia* and most human fungal pathogens remain poorly investigated, the interaction between fungal plant pathogens and their host is much more detailed [7]. Specifically, all organisms must be able to sense their environment and adapt. Animals, plants, and fungi all utilize soluble oxygenated polyunsaturated fatty acids (PUFA) as common signaling molecules. Interestingly, most fungal and plant biologists refer to these signaling molecules as “oxylipins” [8]. However, most human biologists refer to these same structures as “eicosanoids”, which are exclusively derived from 20 carbon chain length arachidonic acid and related PUFA. While structurally similar and likely evolutionarily conserved, the difference in nomenclature has prevented in-depth investigations and cross-discipline collaborations. For the sake of convenience, in this manuscript, we refer to oxylipins and eicosanoids comprehensively as “lipid mediators”.

Skin displays highly active PUFA metabolism, resulting in the production of lipid mediators that modulate both pro- and anti-inflammatory reactions [9]. The activation of phospholipase A2 by solar UVB has been shown to affect fatty acid release, particularly (n)−6 PUFA arachidonic acid (AA; 20:4n−6), and potentially n−3 PUFA eicosapentaenoic acid (EPA; 20:5n−3) [10]. Subsequently, these PUFA are metabolized by cyclooxygenases (COXs), lipoxygenases (LOXs), and cytochrome P450 (CYP-450) and non-enzymatically into a wide range of lipid mediator metabolites [11]. High COX-2, 12-LOX, and 15-LOX activity has been reported in epidermal cells, and infiltrating neutrophils may possess 5-LOX activity [12]. Considerable species differences are reported for LOX expression [13], and additional fatty acid metabolism by cytochrome P450 and non-enzymatic oxidation may contribute to the production of a wide diversity of related lipid metabolites [14].

While these lipid mediators are known to be endogenously generated, recent research indicates that the intestinal microbiota is also capable of participating in lipid metabolism and thus is able to generate bioactive lipid mediators [15]. Recently, fungi not previously known to produce oxylipins have been found to do so, including *Aspergillus fumigatus* [16] and the skin residents *Candida albicans* [17] and *Candida parapsilosis* [18], opening new research vectors on fungal/human communication [19]. As such, research is expanding rapidly examining lipid mediated crosstalk between the human host and the skin microbiota. These explorations would be greatly enhanced by non-invasive methods that can quantitatively assess lipid mediators derived from shared metabolic pathways between skin and its microbial residents.

Advances in mass-spectrometry (MS) have boosted the development of targeted lipidomics, enabling analysis of the complex network of lipid mediators through established experimental procedures [20]. Often, lipid mediator profiles are determined by LCMS in the serum of patients with skin diseases such as psoriatic arthritis and atopic dermatitis to correlate metabolites with disease activity [21,22]. Traditionally, epidermal lipids are collected using non-invasive adhesive tape strips such as Cuderm D’squame^®^ [23]. These tape strips have been optimized to sample and provide information on neutral lipid species found in non-viable stratum corneum [24]. For dermal lipids, the most common sampling strategy is invasive skin biopsy [25]. To our knowledge, currently, there is no reliable, non-invasive technique to collect soluble, non-epidermally bound oxygenated PUFA from the surface of human skin.

Defining the functional role of the skin microbial community in lipid mediator signaling and, hence, its role in skin health and disease will be useful in multiple contexts. Doing so in a quantitative, non-invasive manner will enhance our understanding of skin biochemistry while ensuring minimal subject discomfort and maximal sample collection potential. Therefore, the presented method should be broadly applicable to human/microbe interaction and should enable future research on this highly competitive field.

## 2. Results

### 2.1. Method Development and Validation

The study of skin lipids via commonly available tape strip protocols have issues with high background and absorbance ability limitation [26]. Additionally, as we are primarily interested in signaling lipids of sebaceous and *Malassezia* origin, we designed a strategy to target unbound polar lipids as opposed to membrane bound epidermal lipids that are primarily attached to adhesive tapes. Therefore, we systematically investigated the quantitative differences in skin lipid mediators collected from subjects using different absorbent materials attached to the skin surface (forehead) with adhesive tape. To ensure quantitative collection, we initially considered the lipid background in the paper tape combination, and the extent to which the tape acts as a chromatographic system. Flax paper generally captured the highest absolute amount of relevant lipid mediators (Appendix A). Samples adsorbed to and released from flax paper identified 16 lipid mediators including 3 PUFA successfully detected and quantified from human skin. The blank CuDerm tape gave rise to a noticeable background, but the background of the flax-tape combination was lower than other options. To maximize precision, the flax paper disks were cut into uniform shapes with equal surface areas with a standardized hole punch. To define flax paper absorption and extraction linearity, mixed non-deuterated standards were applied directly to the paper, extracted, and analyzed (Appendix A). Flax paper and solid phase extraction (SPE) recovery rates were sufficient to quantitatively measure lipid mediators isolated directly from human skin (Appendix A).

To maximize lipid mediator collection efficiency from human facial skin, we directly compared lipid levels on 2 different skin sites: the cheeks and the forehead. As shown in Appendix A, all 16 detected lipid mediators and PUFA, except 9,10-diHOME, were more abundant on the cheeks than the forehead. When collecting clinical samples from human volunteers, it is essential to minimize the impact (time) during sampling. Hence, we next investigated the optimal collection time for flax-tape disks to capture lipid mediators on the cheeks for 1, 2, 5, and 10 min. We found a more than adequate sample amount for quantification of lipid mediators with no significant differences between collection time points (Appendix A). We chose 5 min as this time period maximizes the efficiency of simultaneous other clinical activities such as parallel microbiome sampling. Additionally, of relevance when collecting samples from human subjects, it is essential to minimize alteration of the subjects normal routine to best model everyday conditions and to maximize compliance. As the goal was to measure as closely as possible “normal” skin lipid mediator composition, it was decided to minimize the time between the last facial wash and sample collection. In general, collected lipid mediators significantly increased up to 48 h post washing with no significant change when extended to 72 h (Figure 1).

### 2.2. Effect of Washing on Skin Lipid Mediator Profile

As predicted, the levels of lipid mediators detected on skin were drastically reduced by washing (Figure 2A). Importantly, this also indicates that this method is sensitive and linear enough to detect and quantify these changes. Orthogonal partial least square discriminant analysis (OPLS-DA) of the lipidomic data shows a partial inter-group separation between lipid mediator profiles found on skin before and after washing (Figure 2B).

### 2.3. Lipid Mediator Composition Varies between Subject and Site but Shows Temporal Stability

The cheek samples of 10 subjects over a 9-day period were collected during a parallel study to that described by Leong et al. [27]. Lipidomics analysis allowed for the quantification of 30 lipid mediator species, including 4 precursor PUFA (see the Supplementary Excel sheet). We detected weak temporal fluctuation but strong inter-individual differences in all 30 lipid mediator concentrations (Figure 3). The total proportional distribution of lipid mediators revealed inter-individual variability (Figure 3A) with temporal stability within individuals (Figure 3B), similar to previous reports of skin microbiome [28]. Orthogonal partial least square discriminant analysis (OPLS-DA) of the lipidomic data revealed a clear intergroup variance (Figure 3C).

### 2.4. Cultured Malassezia Produce Lipid Mediators Also Found on Human Skin

To assess the production of lipid mediators from commensal *Malassezia* yeasts naturally found on human skin, we analyzed in vitro cultured *Malassezia* cell pellets. Three different *Malassezia* species, *M. globosa*, *M. sympodialis*, and *M. furfur*, were found to produce multiple oxylipin species in vitro (Figure 4). Moreover, the relative oxylipin amounts of the analyzed *Malassezia* species were clearly differentiated from each other, with *M. globosa* showing the highest production (Figure 4A,B). Given that more than half of the detected lipid mediators were also found on human skin (Figure 4C), we hypothesize that separate *Malassezia* species differentially contribute to lipid mediator profiles on human skin.

## 3. Discussion

It is well known that fungal pathogens interact with their plant hosts via PUFA lipid mediators termed oxylipins [29,30]. As very similar oxylipin structures are commonly labelled as eicosanoids in human biology, in this manuscript, we have referred collectively to oxylipins and eicosanoids as “lipid mediators”. As skin resident *Malassezia* are closely related to plant pathogens such as the parasitic *Ustilago* [31], we hypothesize that *Malassezia* would employ similar cross-talk strategies to communicate with skin of their human host. To quantitatively capture and measure lipid mediators derived from both endogenous host and microbial origin, we developed a robust, non-invasive, reproducible sample collection method for downstream LC-MS/MS analysis.

Microbiome-derived bioactive lipid mediators are now believed to modulate the gut-brain axis via bidirectional cross-communication and influence the course of neurodegenerative, metabolic, and gastroenteric diseases [32]. Recent findings expand the influence of the gut microbiome on more distal organs including skin. For example, the linoleic acid-derived 12,13-diHOME produced by neonatal gut bacteria was associated with increased risk of developing asthma and inflammatory skin conditions at later life stages [33]. Gut microbial derived metabolites, including lipid mediators, have been shown to contribute to the development of skin diseases such as atopic dermatitis [34]. The data presented here leads us to hypothesize that the skin microbiome, similar to that in the gut, may affect the skin lipid mediator profile and subsequently skin homeostasis. In agreement with reported weak intra-subject temporal fluctuation of the skin microbiome composition [28], the lipid mediator profile of human facial skin showed significant subject to subject variability with intra-subject temporal stability. While it has been previously reported that dietary PUFA influence skin homeostasis [35], we are not aware of data indicative of the direct impact of skin resident microorganisms on skin lipid mediator profiles. The presented skin lipid mediator analysis tool contribute to a better understanding of the significance of skin microbe metabolic activity on skin surface lipid mediator composition, as accurate, quantitative measurement of skin surface lipid mediators will be key to answer important questions such as whether microbially derived lipid mediators influence skin disease development and progression by modulating host skin inflammation and/or immune responses.

We found that cultured *Malassezia* are capable of producing a rich variety of lipid mediators, many of which are detected on the human skin surface. This suggests that *Malassezia* are at least capable of influencing the human skin lipid mediator profile. Moreover, we detected a strong inter-species variability in lipid mediators produced by the three assessed *Malassezia* species, indicating they may differentially contribute to human skin homeostasis. We cannot conclude with certainty that *Malassezia* directly shape the lipid mediator profile of human skin, given that our lipidomic results were obtained from in vitro cultures and human skin inhabited by *Malassezia*. Further analysis of skin lipid mediators after specific intervention in the skin microflora will give clarity regarding the physiological significance of *Malassezia* and other microbial inhabitants in human skin surface lipid mediator metabolism.

The role of eukaryotic microorganisms in shaping skin lipid mediators and hence skin physiology may be significant, not only because skin resident fungi have been shown to be able to produce lipid mediators per se but also as certain fungi, such as *Malassezia*, are known to be causatively linked to inflammatory skin disorders [36] such as pityriasis versicolor; seborrheic dermatitis; atopic dermatitis; and psoriasis, a chronic immune-mediated disease [37,38,39]. However, the exact role of the microbial community in skin homeostasis remains unclear [40]. It is of note that, in our study, the DHA and AA concentrations appear higher than the metabolites, potentially indicating microbially mediated enzymatic activity affecting overall lipid mediator concentrations.

Our study revealed a common group of lipid mediator species present in both cultured *Malassezia* and on human skin, leading us to hypothesize that these overlapping PUFA metabolites may have fungal origin. Indeed, several overlapping lipid mediator species, such as 8,9-DiHETrE, 5-OxoETE, 9-OxoODE, and 13-OxoODE, were not detectable in human tears [41] or in the supernatant of cultured immortalized N/TERT keratinocytes (results not shown) using identical LC-MS/MS instrument parameters. Interestingly, lipidomic analysis of punch biopsy samples from lesional psoriatic skin showed increased concentrations of individual omega-6 PUFA derived lipid mediators relative to adjacent non-lesional and healthy skin [42]. As highlighted in Figure 4C, all of these elevated lipid mediators were found to be part of the overlapping lipid mediator pool revealed is this study, suggesting that *Malassezia* yeasts may play a role in psoriasis via lipid mediator secretion. In favor of this possibility, the presence of *M. globosa* was found to be increased during episodes of exacerbation of scalp psoriasis [43]. Should future studies show that pro-inflammatory skin microbiome-derived lipid mediators are involved in development or progression of skin disorders, then the respective PUFA metabolizing enzymes could evolve as targets for therapeutic intervention.

Further work will be necessary to investigate linkages between skin PUFA lipid mediators and skin microbial activity, in particular delving into the production of specific lipid mediators by specific microbial species and strains. This will only be accomplished by modulation of the skin microbial community and detailed follow up analyses, hopefully linking this lipid mediator method with transcriptomic profiling. The application of this method will enable deeper investigation of the role of microbially generated PUFA lipid mediators in skin health and disease. We have shown that the absolute quantification of lipid mediators makes LCMS-based skin lipidomics a tool well-suited for rigorous and systematic studies of various topics in skin health and disease, immunology, human skin host–microbe communication, and potentially the impact of drug interventions and cosmetic substances on skin health with respect to their efficacy and claims. Detailed characterization and quantification of the skin microbiome composition at the sub-species level and correlation to total skin lipid mediator concentrations will be crucial for evaluating our hypothesis.

## 4. Materials and Methods

### 4.1. Materials

All endogenous and isotope labelled oxylipin standards were purchased from Cayman Chemical (Ann Arbor, MI, USA). Deionized water was obtained from a MilliQ purification (Millipore, MA, USA). Acetonitrile (MeCN), methanol (MeOH), and Isopropanol (iPrOH) were obtained from Fisher Scientific (Hampton, MA, USA). Acetic acid was obtained from Fisher Scientific (Waltham, MA, USA). Flax paper (Zig-zac, London, UK), lens cleaning paper (Tiffen Lens Cleaning Tissue), Commercially available sebum wipes (Cetaphil, Gentle Skin Cleansing Cloths, MA, USA), D’squame (cat log no: SKU: S100, Cuderm Corporation, Dallas, TX, USA), rice paper (Zig-zac, London, UK), and the Cuderm tapes (CatLog no: SKU-D100, Clinicalandderm, MA USA) used were purchased for testing as the optimal material for sampling of oxygenated PUFA from human skin.

### 4.2. Validation of Lipid Mediator Absorbance Efficiency of Flax Paper

The flax paper was punched into discs with a 1.5 cm diameter using a circle lever puncher. These discs were then transferred onto Cuderm tapes and subsequently used to evaluate the maximum absorbance capacity or to collect samples from skin as illustrated in Appendix A.

In order to ensure that sample collected using flax paper represented an accurate reflection, we assessed the absorbance level of flax paper using a 20 µL mixture of endogenous standards (Appendix A) that was directly spotted onto the flax paper at indicated concentrations (1 ng/mL, 0.5 ng/mL, 0.25 ng/mL, and 0.025 ng/mL). In addition to linearity, the recovery rate of flax paper and solid phase extraction (SPE) efficiency was determined (Appendix A).

### 4.3. Flax Paper Validation as a Sampling Material for Oxygenated PUFA from Skin and Interval Post-Washing Optimization

To examine the suitability and the time needed for the natural skin lipid mediators to reach saturation levels, six healthy volunteers were sampled in accordance with the Declaration of Helsinki and the study was approved by the Bioethical Committee of the National University of Singapore (NUS-B-15-237). These subjects were advised not to shower with known anti-microbial soaps or shampoos for at least 3 days prior to sample collection. Additionally, we applied a strict exclusion criterion for subjects if they had any antibiotic treatments prescribed in the last 6 months.

The experiment was conducted over a period of three weeks. In the first week, each subject was advised not to wash with any skin or hair care product for 72 h, followed by two other weeks with 48 h and 24 h of non-washing prior to sampling. Samples were collected from the cheeks right after washing at four different sampling times, i.e., 1, 2, 5, and 10 min and stored at −80 °C until lipidomics analysis.

### 4.4. Longitudinal Sampling Study to Assess Temporal Intra- and Inter-Individual Differences

In total, 10 healthy subjects aged 21–65 were recruited for the longitudinal study based on the following inclusion criteria: (1) normal skin and scalp, and (2) no active cosmetic product or anti-dandruff containing shampoo use for 2 weeks prior to the study. The subjects were sampled in total six times on cheek, as described above: twice at baseline (days 1 and 2) and four other times (days 3, 4, 7, and 9).

The sampling study was described in detail earlier [27], and reviewed by the National University of Singapore Institutional Review Board (NUS-IRB-15-237). All subjects provided written informed consent prior to study commencement.

### 4.5. Lipid Extraction

The workflow of analysis is illustrated in Appendix A. The lipid mediator extraction was performed based on previous publications [44,45]. To extract lipid mediators from *Malassezia*, lyophilized cell pellets were resuspended in 1 mL water (pH 4.48) followed by homogenization and clearing of the samples by centrifugation. The dried lipid extracts derived from *Malassezia* as well as from skin surface were resuspended in 50 μL of MeCN/water/acetic acid (60/40/0.02, *v*/*v*) prior to MS analysis. (see Appendix A for detail Information).

### 4.6. LC-MS/MS Analysis

For the reverse-phase liquid chromatography (LC), an Agilent 1290 series HPLC system (Agilent, Santa Clara, CA, USA) was employed. The chromatographic separation was performed on Acquity UPLC BEH shield RP18 columns (2.1 × 100 mm; 1.7 m; Waters, MA, USA) [43,44]. Samples were injected with a volume of 10 µL. The LC was coupled to a Agilent 6495 triple quadrupole (QQQ) mass spectrometer (Agilent, Santa Clara, CA, USA). (see Appendix A for detail Information).

### 4.7. Statistical Analysis

The data were analyzed using GraphPad Prism, version 6 for windows software (GraphPad Software, La Jolla, CA, USA), and some analysis partially were carried out using Microsoft Excel version 2013. The differences in between the group studies were examined using independent student’s T-test. Heat map clustering assays and Orthogonal partial least square discriminant analysis (OPLS-DA) were performed using MetaboAnalyst 5.0 [46].

## Figures and Tables

**Figure 1 metabolites-11-00700-f001:**
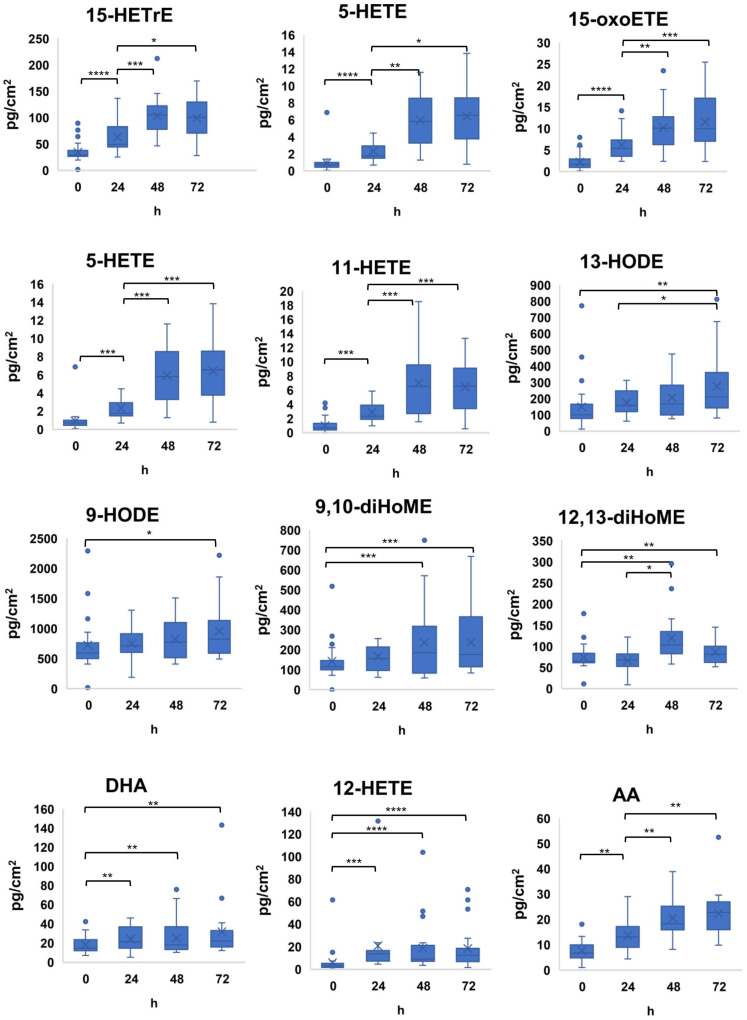
The results using scatter dot plots showing the median (bold horizontal line), interquartile range (box), and lipid mediator concentrations in the Y-axis across four different pre-selected time intervals (0, 24, 48, and 72 h without washing). The results are the mean ± S.E. *, *p* ≤ 0.05 **, *p* ≤ 0.01, *** *p* ≤ 0.001, and **** *p* ≤ 0.0001. h refers to hours.

**Figure 2 metabolites-11-00700-f002:**
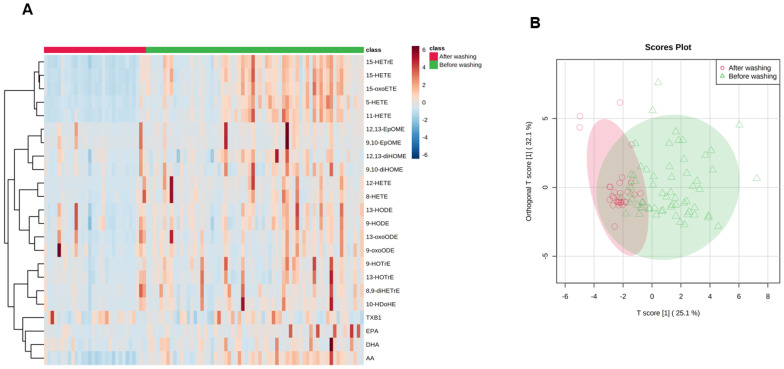
Heat map clustering and orthogonal partial least square discriminant analysis (OPLS-DA) comparing lipid mediator concentrations produced on skin before and after washing. (**A**) Heat map clustering analysis using Ward’s algorithm of skin lipid mediator concentrations that differed significantly between pre- and post-wash conditions, based on one-way ANOVA tests. The color scale from −6 to 6 represents the z-score, with positive z-score values in red indicating increased lipid mediator concentrations and negative z-score values in blue reflecting decreased lipid mediator concentrations before and after washing. (**B**) OPLS-DA comparing lipid mediator concentrations before and after washing.

**Figure 3 metabolites-11-00700-f003:**
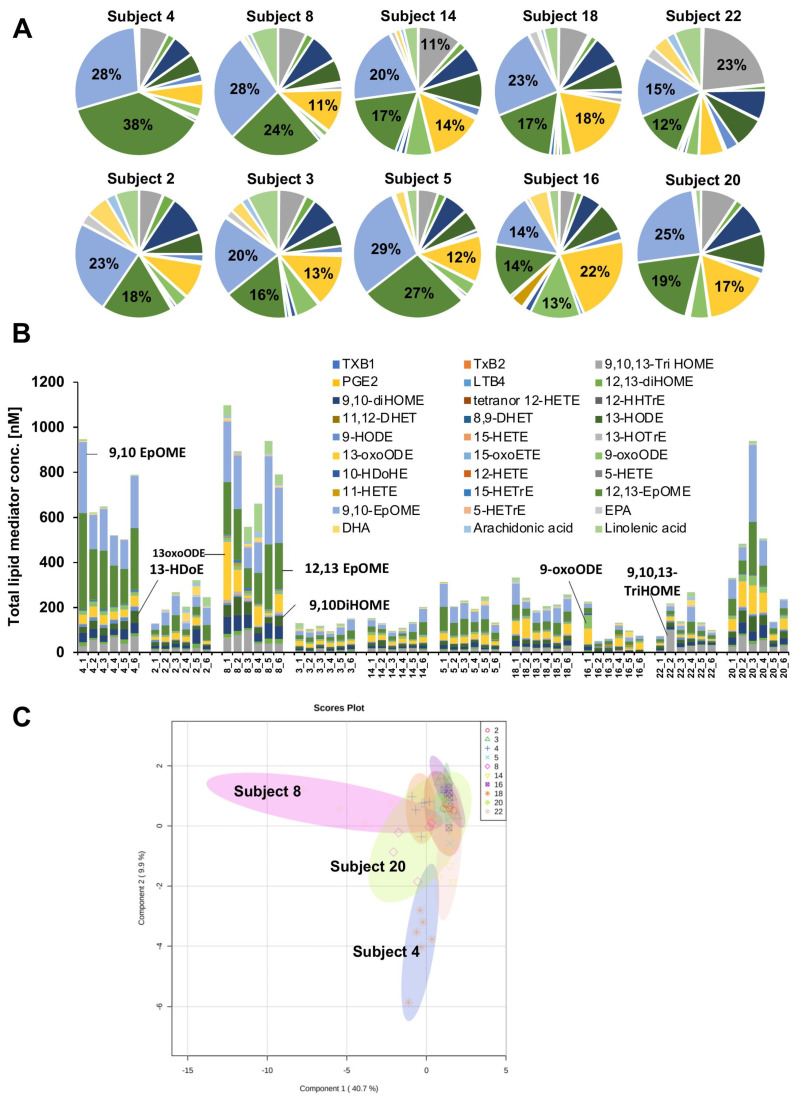
Comparative quantification of lipid mediators in the cheek samples by targeted lipidomics. (**A**) Pie charts showing total lipid mediators detected on the skin of individual subjects and their proportional distribution. The percentage of lipid mediators is only indicated for species that are represented by at least 10%. (**B**) Stacked bar charts show concentrations of lipid mediators that were quantified on the indicated 6 consecutive days, grouped by subjects. (**C**) sPLS-DA comparing lipid mediator concentrations between different subjects. 95% confidence region is shown for each sample group. Only the most distinct subjects are labeled.

**Figure 4 metabolites-11-00700-f004:**
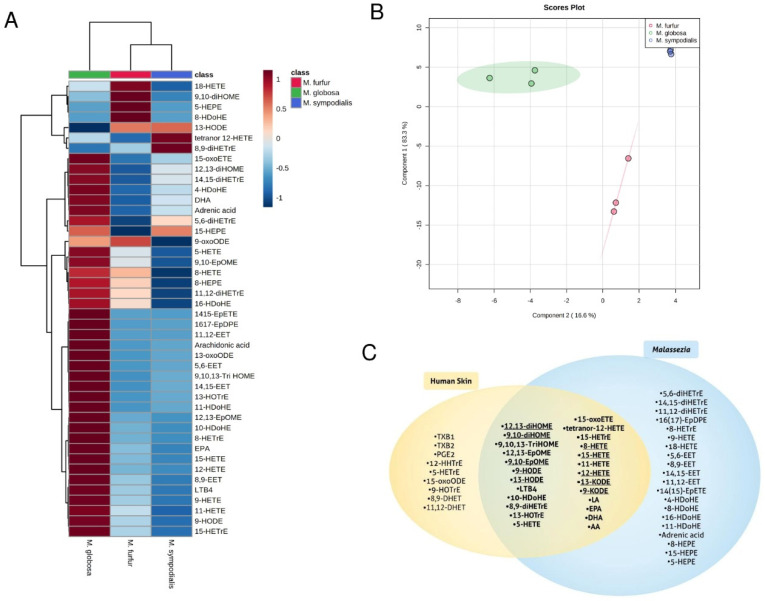
*Malassezia* are capable of producing lipid mediators that can be detected on human skin. (**A**) Heat map clustering analysis using Ward’s algorithm of skin lipid mediator concentration averages from triplicate data sets that differed significantly between three different *Malassezia* species, based on one-way ANOVA tests. (**B**) OPLS-DA comparing lipid mediator concentrations between three different *Malassezia* species. (**C**) Venn diagram of overlapping lipid mediators between human skin and *Malassezia.* Underlined lipid mediators were detected in psoriatic skin (38). Total 54 different lipid mediator species, of which 25 are overlapping between human skin and *Malassezia*.

## Data Availability

The data presented in this study are available in Appendix A.

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
