# Peer review of "Host/Malassezia Interaction: A Quantitative, Non-Invasive Method Profiling Oxylipin Production Associates Human Skin Eicosanoids with Malassezia"

_metabolites, 2021, doi:10.3390/metabo11100700_

Round 1
Reviewer 1 Report
In this manuscript, the authors investigated the method profiling skin eicosanoids and the roles of Malassezia on skin eicosanoid production. I also think that this topic may be of interest to the field and the experimental strategies are well designed. The manuscript is well written, and the results are clearly presented. I have a few minor comments, explained blow.
In Figure 1, it might be preferable to change the unit of Y-axis from nM to pg/cm2. The authors should describe the detailed MS/MS condition (parent m/z, product m/z, collision energy, etc.) of all compounds and the internal standards. In general, the concentration of DHA and AA is higher than the metabolites. The authors should check the concentrations of DHA and AA.
In Figure 3, it should be possible to determine which color is which compound.
Author Response
Reviewer #1
In this manuscript, the authors investigated the method profiling skin eicosanoids and the roles of Malassezia on skin eicosanoid production. I also think that this topic may be of interest to the field and the experimental strategies are well designed. The manuscript is well written, and the results are clearly presented. I have a few minor comments, explained blow.
In Figure 1, it might be preferable to change the unit of Y-axis from nM to pg/cm2. The authors should describe the detailed MS/MS condition (parent m/z, product m/z, collision energy, etc.) of all compounds and the internal standards.
In general, the concentration of DHA and AA is higher than the metabolites. The authors should check the concentrations of DHA and AA.
Thank you for the useful comments. We updated the Y-axis unit to pg/cm2.
The detailed MS/MS conditions (parent m/z, product m/z, collision energy, etc.) of all compounds and the internal standards are included in the current supplementary file (Table S1&S2).
Yes, we confirm that the DHA and AA concentrations are higher than the metabolites, which may be due to the influence of the skin microbiome. We added a sentence referencing this observation on page 15, line 19 of the discussion.
In Figure 3, it should be possible to determine which color is which compound.
After having tested all possible different colour combinations, we came to the conclusion that the current colour choice is best suited to distinguish between 30 different colour tones and lipid mediator species, respectively. We have avoided the use of red, due to the prevalence of red/green colour visualization issues. For a better understanding, we now indicated the names of the most prevalent lipid mediator species.
We have also included additional supplemental information by adding an Excel sheet presenting all lipid mediator concentrations relevant to Figure 3 (see supplementary Excel sheet – Temporal Analysis).
Reviewer 2 Report
Ambaw et al. proposed a study in which they developed a highly sensitive mass-spectrometry based method to quantify 23 bioactive polyunsaturated fatty acids from the human skin surface and in vitro Malassezia cultures.
I do believe that the authors developed a trustable and efficient LC-MS/MS method for the estimation of eicosanoids in human skin. The results are clear, supported by robust experimental evidence and deeply discussed. Moreover, the supplementary materials are adequate and properly explained. The main limitation of the present study is disclosed by the authors themselves in line 268 “Detailed characterization and quantification of the skin microbiome composition at the sub-species level and correlation to total skin lipid mediator concentrations will be crucial for evaluating our hypothesis”. However, I do believe that studying the human skin microbiota is essential to hypothesize the origin of the lipid found on the skin. How can the authors address this point without compromising the primary goal of the study itself? The significance of this study without these experiments is severely compromised.
I would consider a major revision of the paper.
Some concerns:
- Please, the authors should consider shortening the title, which to me seems to be wordy
- Figure 1. The graphs of the concentrations of 13-HODE and 9-HODE should start with 0. There is no scientific explanations to start an absolute concentration graph with negative values.
- The heatmap in Figure 2 was not ordered by groups in order to highlight the clustering algorithm, however, for the reader is very difficult to understand the general pathway between the lipid mediator produced on skin before and after washing. I do recommend to the authors to avoid the clustering favoring the group separation.
- Figure 3. The color choice should be reconsidered. It is quite impossible to distinguish between the different tones of either yellow, green and blue.
- Material and methods lack a proper description of in vitro Malassezia cultures.
Author Response
Reviewer #2
Ambaw et al. proposed a study in which they developed a highly sensitive mass-spectrometry based method to quantify 23 bioactive polyunsaturated fatty acids from the human skin surface and in vitro Malassezia cultures.
I do believe that the authors developed a trustable and efficient LC-MS/MS method for the estimation of eicosanoids in human skin. The results are clear, supported by robust experimental evidence and deeply discussed. Moreover, the supplementary materials are adequate and properly explained. The main limitation of the present study is disclosed by the authors themselves in line 268 “Detailed characterization and quantification of the skin microbiome composition at the sub-species level and correlation to total skin lipid mediator concentrations will be crucial for evaluating our hypothesis”. However, I do believe that studying the human skin microbiota is essential to hypothesize the origin of the lipid found on the skin. How can the authors address this point without compromising the primary goal of the study itself? The significance of this study without these experiments is severely compromised.
I would consider a major revision of the paper.
Some concerns:
- Please, the authors should consider shortening the title, which to me seems to be wordy
Thank you for the insightful feedback. We have made multiple adjustments to the manuscript at your suggestion, including shortening and focusing the title (listed in detail below).
To address your overall concern, that ideally this study should link the production of skin lipid mediators to sub-species level microbial analysis. We could not agree more, this type of data would be the Rosetta Stone of skin, and indeed any, microbiome research. However, we respectfully submit that as all human skin is populated by a complex microbial community, it will be many years, if not decades, of research to try and dissect the roles of specific species and strains. We can imagine no way to complete this task other than perturbing the microbial community with species or strain specific antimicrobials, which do not exist today. Even specific removal of kingdoms, such as fungi versus bacteria, would require significant effort, clinical resources, and time, and we believe this research lies beyond the scope of this manuscript. Our hope is that this methodology, results, and discussion will enable and inspire work of the type mentioned which we both look forward to in future research.
To this end, we have added a sentence to this effect on page 16, line 14, of the discussion.
- Figure 1. The graphs of the concentrations of 13-HODE and 9-HODE should start with 0. There is no scientific explanations to start an absolute concentration graph with negative values.
We apologize for the presentation error, thank you for the comments. We have checked the calculations and corrected the negative values.
- The heatmap in Figure 2 was not ordered by groups in order to highlight the clustering algorithm, however, for the reader is very difficult to understand the general pathway between the lipid mediator produced on skin before and after washing. I do recommend to the authors to avoid the clustering favoring the group separation.
Thank you for the thoughtful idea. As suggested, we created a heatmap avoiding group clustering. However, in this map segregation of low versus high concentrations of the respective lipid mediator species becomes visually less obvious, despite the PCA indicating a clear separation of the two groups (Fig. 2B). We therefore conclude that the unsupervised heatmap presented in Figure 2A is best suited to visualize the clustering of similar compounds, such as e.g. the HETE’s, the diHOME’s and EpOME’s, the HODE’s and KODE’s, in the same region and the same subgroup, thus reflecting their similar functionality.
- Figure 3. The color choice should be reconsidered. It is quite impossible to distinguish between the different tones of either yellow, green and blue.
As also mentioned above, we tested all many different colour combinations and came to the conclusion that the current colour choice is the best we can come up with to distinguish between 30 different colour tones and lipid mediator species, respectively. To note, we have avoided the color red due to the prevalence of difficulty visualizing red and green. We will also consult with the journal editorial department for input.
To improve reader understanding we indicated the names of the most prevalent lipid mediator species in the figure.
Furthermore, we also included additional supplemental information by adding an Excel sheet presenting all lipid mediator concentrations relevant to Figure 3 (see supplementary Excel sheet – Temporal Analysis).
- Material and methods lack a proper description of in vitro Malassezia cultures.
Please see Appendix S1: Analysis of lipid mediators from cultured Malassezia. A more detailed description of the culturing conditions and reference are added.
Reviewer 3 Report
The manuscript by Abmaß et al. describes an approach to investigate the role of oxylipins via a non-invasive methodology profiling skin eicosanoids. This is a novel study showing the potential of high sensitive mass spec analytics to find lipid pattern associated to microbial communities on the skin surface and the potential impact on the human being. Overall the study is well performed, provides valuable new data and paves the way for additional studies on the role of oxylipins on the skin microflora and the human health. Furthermore it is the first method for the non-invasive analytics of skin associated lipid mediators via mass spectrometry.
There are some points to improve:
1) references: it seems that no pages in the references are present; maybe this is an error of the citing program; but it should be corrected in a revision
2) software versions: please provide information about the versions of the software used; in particular for MetaboAnalyst since it will be updated regularly
3) lipid mediator analytics: I could not find the composition of the deuterated standard mix used; please add this information
4) ACN vs MeCN: I know it is a typical abbreviation; but in terms of "near IUPAC" nomenclature consider using MeCN instead of ACN - not al all critical; since it is used widely!
5) The same goes with IPA; here I suggest iPrOH
6) Please change "OPLSDA" to "OPLS-DA" which is the more commonly used term
7) There are some formatting errors (missing or additional specs) and typos which should be corrected thorough text editing
Author Response
Reviewer #3
The manuscript by Abmaß et al. describes an approach to investigate the role of oxylipins via a non-invasive methodology profiling skin eicosanoids. This is a novel study showing the potential of high sensitive mass spec analytics to find lipid pattern associated to microbial communities on the skin surface and the potential impact on the human being. Overall the study is well performed, provides valuable new data and paves the way for additional studies on the role of oxylipins on the skin microflora and the human health. Furthermore it is the first method for the non-invasive analytics of skin associated lipid mediators via mass spectrometry.
Thank you for the kind review and useful comments. We think we have addressed them all below.
There are some points to improve:
1) references: it seems that no pages in the references are present; maybe this is an error of the citing program; but it should be corrected in a revision.
Page numbers have been added where possible.
2) software versions: please provide information about the versions of the software used; in particular for MetaboAnalyst since it will be updated regularly.
Reference for Metaboanalyst has been updated to the used newest version.
3) lipid mediator analytics: I could not find the composition of the deuterated standard mix used; please add this information.
Thank you for the comments. We included the composition of all deuterated standards and their concentrations in Table S2.
4) ACN vs MeCN: I know it is a typical abbreviation; but in terms of "near IUPAC" nomenclature consider using MeCN instead of ACN - not al all critical; since it is used widely!
ACN has been changed to MeCN.
5) The same goes with IPA; here I suggest iPrOH.
IPA has been changed to iPrOH.
6) Please change "OPLSDA" to "OPLS-DA" which is the more commonly used term.
OPLSDA has been changed to OPLS-DA.
7) There are some formatting errors (missing or additional specs) and typos which should be corrected thorough text editing
We have carefully reread the manuscript and made a number of modifications based on this feedback.
Round 2
Reviewer 2 Report
Essentially the authors addressed all my main concerns. Even if they cannot link the production of skin lipid mediators to microbiota directly, the paper could be accepted in the present form.